# Profiling Elastoplastic and Chemical Parameters to Assess Polymerization Quality in Flowable Bulk-Fill Composites

**DOI:** 10.3390/bioengineering11020159

**Published:** 2024-02-05

**Authors:** Nicoleta Ilie

**Affiliations:** Department of Conservative Dentistry and Periodontology, University Hospital, Ludwig-Maximilians-University, Goethestr. 70, D-80336 Munich, Germany; nilie@dent.med.uni-muenchen.de

**Keywords:** bulk-fill composite, degree of cure, depth of cure, hardness, modulus of elasticity, creep, strength, fractography, reliability

## Abstract

In the chronology of polymer-based composite materials, flowable bulk-fill composites represent the most recent development. They enable a significant reduction in treatment time by being applied in larger increments of 4 to 5 mm. The aim of the investigation was to assess the polymerization quality and mechanical performance of a new formulation that has just entered the market and was still in experimental formulation at the time of the investigation, and to compare these results in the context of clinically established materials of the same category. Adequate curing in increments of up to 4 mm could be confirmed both by profiling the elastoplastic material behavior of large increments in 100 µm steps and by real-time assessment of the degree of conversion and the associated polymerization kinetic. A slightly lower amount of filler in the experimental material was associated with slightly lower hardness and elastic modulus parameters, but the creep was similar and the elastic and total indentation work was higher. The kinetic parameters were assigned to the specific characteristics of each tested material. The mechanical macroscopic strength, evaluated in a three-point bending test and supplemented by a reliability analysis, met or exceeded the standards and values measured in clinically established materials, which for all materials is related to the higher flexibility of the beams during testing, while the modulus of elasticity was low. The low elastic modulus of all flowable bulk-fill materials must be taken into account when deciding the clinical indication of this material category.

## 1. Introduction

The simplification of restoration techniques when using polymer-based composites has led to the now generally accepted [1] and even advocated bulk filling of deeper cavities. Although in bulk filling the use of increments should not exceed 4 mm (5 mm for few materials) even when a cavity is larger, it consistently speeds up the restoration process [2] and reduces the risk of contamination with oral fluids or the inclusion of defects between layers. Based on these considerations, bulk-fill composites are marketed as materials that are particularly suitable for patients with limited compliance.

A general distinction is made between low-viscosity (flowable) and high-viscosity materials, based on the different viscosity and application technology. Since the viscosity is primarily influenced by the proportion of inorganic fillers, flowable bulk-fill composites have a lower proportion and therefore implicitly poorer mechanical properties compared to highly viscous materials [3]. Nevertheless, their good flow properties and handling make them very popular in clinical use, as they are able to fill even a small gap and give the appearance to fit better on the cavity walls. In vitro studies are somehow sobering in this regard as they have shown a higher imperfection margin percentage and polymerization shrinkage stress compared to highly viscous bulk-fill and hybrid composites [4]. This statement needs to be put into perspective somewhat, as internal adaptation and bond strength are connected to the cavity configuration factor (C-factor), with the incremental technique evidencing higher bond strength than the bulk-fill technique in cavities with a high C-factor, while in cavities with a low C-factor the bond strength was similar [5]. When subjecting class II cavities to degradation, in vitro studies suggest similar behavior of the bulk and incremental filling technique for dentin, but the enamel margin fit was improved when using a higher modulus material placed incrementally [6]. Another study is very cautious about the use of bulk-fill resin composites with high shrinkage strain after evaluating the interfacial integrity of deep (6 mm) Class-II (OM/OD) restorations [7]. Likewise, high-viscosity bulk-fill restorations, but not flowable bulk-fill restorations, demonstrated similar marginal integrity to traditional hybrid composite restorations after thermal aging in both deciduous and permanent molars [8]. Further, when comparing the clinical effectiveness of resin-based direct posterior restorations in a recent meta-analysis, no statistically significant difference was found between hybrid, micro-hybrid, nano-hybrid, and bulk-fill resin composites in terms of color matching, surface finish, material fractures, and anatomical shape [9].

In terms of chemical composition, bulk-fill and conventional composite materials are largely identical. In addition, all material development trends in recent years have been implemented in bulk-fill composite materials, including nanotechnology for fillers [10], bioactive filler systems [10,11], new polymerization mechanisms such as RAFT (reversible addition fragmentation chain transfer polymerization) [12], Ormocer (acronym for organically modified ceramics) polymer matrices [13], or novel photo-initiator systems [14]. These facts raise the fundamental question of why bulk-fill composites can be polymerized in four-to-five-millimeter increments, while traditional composites must be layered in much thinner, two-millimeter increments. The answer essentially lies in the higher translucency (light transmission ability) of the bulk-fill composites, which depends on the material composition [15], such as differences in the refraction index of the filler and monomer matrix [16], and the microstructure. In fact, low-viscosity bulk-fill composites have been shown to be more translucent than high-viscosity bulk-fill composites, which in turn are more translucent than traditional composites. In this context, translucency means the light transmission ability of the wavelengths of light from a light curing unit (LCU) that are required to initiate the photo-initiators.

While the increased translucency was originally achieved essentially by increasing the filler size and decreasing the amount of filler to reduce the filler–matrix interface where scattering occurs, or by decreasing the amount of pigments and opacifier, bulk-fill polymer-based composites have evolved into modern, complex materials that no longer rely on translucency to allow sufficient curing in the depth. The use of reversible-addition-fragmentation-chain-transfer (RAFT)-mediated polymerization, for instance, reduces the need for high translucency and has been successfully implemented in two commercially available [17] dental composites [18]. Such developments are of great importance for improving the aesthetics of bulk-fill composite restorations, as highly translucent materials placed in large cavities may not adequately mask tooth discoloration or may appear grayish.

Despite the innovations presented above, applying a composite in layers of up to 4–5 mm entails the increasing risk that not enough light will reach the bottom of the restoration and therefore insufficient polymerization may occur in deeper layers. This can lead to a number of deficiencies in the material such as a low degree of monomer conversion [19] that induces elution of the unreacted monomers [19] associated with increased toxicity [20] and hypersensitivity and lower mechanical and chemical stability. All of these aspects can affect the patient’s health and the longevity of the restoration.

As the development of novel materials advances at a rapid pace, with additionally skills and the aim of versatility and universal application, the need for careful and thorough material characterization from physicochemical and mechanical perspectives must be emphasized. The aim of the present study was therefore to analyze a novel, experimental flowable bulk-fill composite formulation with enhanced opacity in terms of its sufficiency of curing under clinically relevant polymerization conditions, and its elastoplastic micro- and macroscopic behavior with respect to clinically confirmed materials.

The null hypotheses tested state that the experimental flowable bulk-fill composite formulation behaves similarly to established materials with respect to (a) the degree of cure and kinetic of polymerization at various depths; (b) the depth of cure and variation in elastoplastic behavior profiled across the depth of large, simulated restorations; (c) macro-mechanical behavior, fracture behavior, and reliability.

## 2. Materials and Methods

### 2.1. Materials

An experimental flowable bulk-fill RBC (Exp) was evaluated in comparison to two commercially available materials (Table 1) by evaluating a range of physical, mechanical, and chemical parameters to describe its behavior during light exposure and post-curing.

### 2.2. Methods

Micro-mechanical properties profiling was performed in 100 µm from top to bottom of 6 mm specimens. Variations in the positioning of the profile, either in the center or at peripheral locations, were tracked. Depth of cure (DOC) was assessed 24 h post-curing from micro-indentation profiling of the Vickers hardness. Furthermore, the degree of cure was assessed in real time at 2 mm and 4 mm depths during light exposure, which was 20 s in all materials, and up to 5 min post-cure. In addition, the mechanical properties were determined on a macroscopic scale in a three-point bending test and were supplemented by fractography, reliability analysis, as well as light and SEM microscopy. The light curing device (LCU) used was characterized spectrophotometrically in order to determine the emission spectrum, the incident irradiance, and the constancy of the parameters during the exposure period both at direct contact and at exposure distances of up to 10 mm.

#### 2.2.1. Spectrophotometry: LCU Characteristics

The LCU characteristics were assessed on a laboratory-grade USB4000 spectrometer (MARC (Managing Accurate Resin Curing) System, Blue light Analytics Inc., Halifax, NS, Canada) referenced by the National Institute of Standards and Technology (NIST). The spectrometer employs a 3648-element Toshiba linear Charge-coupled Device (CCD) array detector and high-speed electronics (Ocean optic, Largo, FL, USA) and was calibrated using an Ocean Optics’ NIST-traceable light source (300–1050 nm). The system uses a CC3-UV Cosine Corrector (Ocean optic, Largo, FL, USA) to collect radiation over a 180° field of view, thus mitigating the effects of optical interference associated with light collection sampling geometry.

Irradiance was determined on five occasions by placing the LCU centered and perpendicular to the spectrophotometer sensor (3.9 mm diameter), while varying the exposure distance in 1 mm steps from direct contact to a distance of 10 mm. The irradiance recorded by placing the LCU directly on the sensor represents the incident irradiance, i.e., the irradiance that hits the sample surface when curing the specimens. Irradiances in a wavelength range of 360–540 nm were recorded individually at a rate of 16 recordings/s. The sensor was triggered at 20 mW. The exposure time was 20 s. 

#### 2.2.2. Instrumented Indentation Test (IIT): Quasi-Static Approach (ISO 14577 [21]) to Determine Micro-Mechanical Properties Profiles and Depth of Cure (DOC)

Cylindrical specimens (*n* = 6; height = 6 mm; diameter = 10 mm) were prepared in cylindrical metal molds. Specimens were cured for 20 s by applying the LCU perpendicular, directly on the top surface of the cylinder. Specimens were then removed from the mold and stored in 37 °C distilled water for 24 h. Prior to measurement, specimens were sectioned in the middle, parallel to the direction of curing, with a slow-speed diamond saw (Isomet low-speed saw, Buehler, Germany) under water cooling. Each sample was then fixed to a glass slide, mounted in an automatic grinding machine (EXAKT 400CS Micro Grinding System, EXAKT Technologies Inc., Oklahoma City, OK, USA), wet-ground with silicon carbide sand paper (grit size p2500 and p4000, LECO Corporation, St. Joseph, MI, USA), and polished with a diamond suspension (mean grain size: 1 µm). Measurements were made with an automatic micro-hardness indenter (FISCHERSCOPE^®^ HM2000, Helmut Fischer, Sindelfingen, Germany) starting at 0.1 mm under the surface, with 100 µm intervals between the measuring points, summarizing 360 single indentations for each material.

Indentation depth was monitored simultaneously with the indentation force during an indentation cycle representing an increase in indentation force within 20 s from 0.4 mN to 1000 mN, a holding time of 5 s at maximum force to assess creep (Cr), and a subsequent reduction in force within 20 s at a constant speed. The integral of the indentation force over the depth outlines the total mechanical work W_total_ (=∫Fdh). This is partially consumed as plastic deformation work W_plastic_, while the rest is released as work of the elastic recovery W_elastic_. The indentation modulus (E_IT_) was calculated from the slope of the tangent of the indentation depth curve at maximum force. Further parameters, such as hardness, were calculated by evaluating the impression created during indentation. The resistance to plastic deformation is expressed by the Vickers hardness (HV) using an indenter correction based on the Oliver and Pharr model and described in ISO 14577 [21] and a previous calibration with sapphire and quartz glass. Plastic and elastic deformation is expressed by the universal hardness (or Martens hardness, HM).

The depth of cure (DOC), acknowledged in composites as the thickness that is adequately cured, was defined arbitrarily as the depth where HV equaled 80% of the surface value.

#### 2.2.3. Degree of Conversion (DC)

DC was measured using Fourier transform infrared (FTIR) spectroscopy with an FTIR spectrometer fitted with an attenuated total reflectance (ATR) accessory (Nicolet iS50, Thermo Fisher, Madison, WI, USA). Profiling was performed in real time over 5 min taking one spectra each 0.4 s using two distinct specimen geometries at specimen depths of 2 and 4 mm. Teflon specimen molds (internal dimensions: 3 mm diameter, 2 and 4 mm-thick) were filled in bulk with the RBCs under investigation. For the investigated groups (two specimen depths, three RBCs), 6 samples were employed to determine DC. The non-polymerized material was applied directly onto the diamond ATR crystal in the respective molds and covered with a transparent matrix strip. All materials were irradiated for 20 s.

DC was calculated by assessing the variation in peak height ratio of the absorbance intensities of methacrylate carbon to carbon (C=C) double bond peak at 1634 cm^−1^ by employing the aromatic C=C double bond peak at 1608 cm^−1^ as an internal standard during polymerization of the uncured material using Equation (1):(1)$${\mathrm{DC}}_{\mathrm{Peak}}\%=[1-\frac{{(1634\mathrm{cm}}^{-1}{/1608\mathrm{cm}}^{-1}{)\;}_{\mathrm{Peak}\;\mathrm{height}\;\mathrm{after}\;\mathrm{curing}}}{{(1634\mathrm{cm}}^{-1}{/1608\mathrm{cm}}^{-1}{)\;}_{\mathrm{Peak}\;\mathrm{height}\;\mathrm{before}\;\mathrm{curing}}}]\times 100$$

For each material, the increase in DC with time was described by an exponential sum function as outlined in Equation (2) and previously described in detail in [22]
(2)$$y=a\times (1-e^{-bx})+c\times (1-e^{-dx})$$
where the parameters *a*, *b*, *c*, and *d* were the modulation factors of the exponential function to optimize the exponential function on the measured curve plotted on a DC versus time curve.

#### 2.2.4. Three-Point Bending Test

In total, 60 (n = 20) specimens (2 mm × 2 mm × 18 mm) were prepared in a white polyoxymethylene mold according to the recommendation of ISO 4049:2019 [23] for the three-point bending test. Light exposure followed the protocol specified in the standard, which included irradiation on the top and bottom of the samples for 20 s, with three light exposures overlapping an irradiated section by no more than 1 mm of the light guide diameter to prevent multiple polymerizations. Immediately after light exposure, specimens were removed from the mold and were ground with silicon carbide paper (P 1200 grit, LECO Corporation) to eliminate interfering edges or bulges, and then immersed in distilled water at 37 °C for 24 h in a dark environment. The flexural strength (FS), flexural modulus (E), and beam deflection (ε) were determined in a three-point bending test according to NIST No. 4877 and considering a span of 12 mm. Therefore, samples were loaded in a universal testing machine until fracture (Z 2.5 Zwick/Roell, Ulm, Germany) at a crosshead speed of 0.5 mm/min. The force in bending was measured as a function of beam deflection, and the slope of the linear part of this curve was used to calculate the flexural modulus.

#### 2.2.5. Light and Scanning Electron Microscopy (SEM) Characterization

All fractured surfaces of the specimens tested in the three-point-bending test were analyzed under a stereomicroscope (Stemi 508, Carl Zeiss AG, Oberkochen, Germany) to determine the fracture pattern and origin of fracture, and were imaged with a microscope extension camera (Axiocam 305 color, Carl Zeiss AG, Oberkochen, Germany). Fractures have been found to originate from either volume (below the surface) or surface (edges and corners) defects. Further, the microstructure of the analyzed materials was assessed by scanning electron microscopy (SEM, Zeiss Supra 55 V P, Carl Zeiss AG, Oberkochen, Germany) on samples that were prepared similar as above (n = 3) and wet-processed using an automatic grinder (EXAKT 400CS Micro Grinding System, EXAKT Technologies Inc., Oklahoma City, OK, USA) with gradually finer silicon carbide abrasive papers (1200, 1500, 2000, and 2400 grit). Surface preparation was completed by polishing the surface with 1 µm diamond spray (DP-Spray, STRUERS GmbH, Puch, Austria).

### 2.3. Statistical Analyses

The distribution of the variables was tested using the Shapiro–Wilk method and allowed for a parametric approach to be used. Multifactor analysis of variance was applied to compare the parameters of interest (flexural strength; flexural modulus; beam deflexion; Martens, Vickers, and indentation hardness; elastic and total indentation work; creep; indentation depth; DOC, DC). The results were compared using one- and multiple-way analysis of variance (ANOVA) and the Tukey honestly significant difference (HSD) post-hoc test using an alpha risk set at 5%. A multivariate analysis (general linear model) assessed the effect of the main parameters and their combinations. The partial eta-squared statistic reported the practical significance of each term based on the ratio of the variation attributed to the effect (SPSS Inc. Version 27.0, Chicago, IL, USA).

Flexural strength data were additionally analyzed by Weibull statistics [24]:$$P_{f}(\sigma_{c})=1-\exp\left[-{\left(\frac{\sigma_{c}}{\sigma_{0}}\right)}^{m}\right]$$
where $\sigma_{c}$ is the measured strength, m the Weibull modulus, and $\sigma_{0}$ the characteristic strength, defined as the uniform stress at which the probability of failure is 0.63 [24].

## 3. Results

### 3.1. Light Curing Unit (LCU) Characteristics

The LCU used for polymerization was a blue LED (Coltolux^®^ LED, Coltene Whaledent Inc., Cuyahoga Falls, OH, USA) with a spectral distribution in the blue wavelength range from 410 nm to 490 nm and a peak at 444.2 nm (Figure 1a). Maximal irradiance was (1973.9 ± 16.6) mW/cm^2^ when the LCU was placed centrally and directly above the sensor (exposure distance = 0 mm). This value represents the incident irradiance to the surface of the composite specimens. The corresponding radiant exposure at the 20 s exposure duration was (39.7 ± 0.4) J/cm^2^. For each exposure distance (Figure 1b), the irradiance reached the maximum value within 0.3 s after turning on the LCU and remained constant throughout the entire exposure duration (20 s). The irradiance varied with exposure distance following an exponential decay (Figure 1c), which was well fitted (R^2^ = 0.997) by a three-parameter equation (f = y_0_ + a × exp(−b × x); with y_0_ = 848.3; a = 1134.6 and b = 0.2).

### 3.2. Instrumented Indentation Test (IIT): Micro-Mechanical Parameter Profiling and Depth of Cure

The variation in the measured parameters in the depth of the 6 mm test specimen is shown below for the three different positions of the 10 mm-wide section. For this purpose, a central position line (central 5 mm) was approached as well as two peripheral positions, each positioned 3 mm to the left (peripheral 2 mm) and right (peripheral 8 mm) in relation to the central (central 5 mm) position. The specified mm corresponds to the 10 mm-wide test specimen, with 0 mm being the position located at one margin and 10 mm the other. Measurements were recorded in 0.1 mm steps.

The variation in HV, HM, E_IT_, and Cr for the experimental composite (Exp) as a function of sample depth and measurement position is summarized in Figure 2a–d. The variation in the measured parameters was non-linear, with a slight increase in properties up to a depth of 1–2 mm, followed by a constant variation and the beginning of a slight decrease starting with a depth of 4 mm. There was a very slight superiority of the central position over the peripheral positions, while both peripheral positions appeared to be identical. The opposite of the described behavior was observed for creep, where there were no differences between measurement positions. Creep values began to increase at depths around 4 mm.

The measured parameters in SDR+ were higher compared to Exp (see individual statistics below) but followed a similar pattern of variation with depth. The curvature of variation with depth just below the surface was much more pronounced compared to Exp (Figure 3).

The measured values for TPF were very close to SDR+ and were higher compared to the experimental composite (see statistics below), while the pattern of variation with the depth and width of the samples was as described above (Figure 4a–d).

In the following, the materials are compared directly in order to make the differences more visible, for which the data measured at the three different positions (central and two peripheral) are pooled together. In addition to the above parameters, the parameters n_IT_, We, and W_t_ are given. For each parameter, one-way ANOVA with subsequent Tukey’s post-hoc test was performed at depths of 0.1 mm, 2 mm, and 4 mm (Table 2). Data indicated that the creep was identical for all materials. For the parameters HM, HV, and E_IT_, SDR+ and TPF performed significantly better than Exp in most comparisons. Exp showed a statistically significantly higher elastic and total indentation work, while for n_IT_, the ratio between W_e_/W_t_ was also higher.

Figure 5 summarizes the direct comparison of the measured parameters for all analyzed materials.

Furthermore, the data were analyzed by multifactorial analysis to determine the effect strength of the main parameters and their interaction with the measured properties. Except for one parameter (n_IT_ was not influenced by the parameter position), the effects of the fixed parameters—material, filler volume %, position, and depth—were all significant (*p* < 0.001). The effect strength was quantified by the partial eta-squared values and is summarized in Table 3. The material itself, followed by the filler volume percentage, showed the significant strongest influence on all measured parameters (higher partial eta-squared values), except for creep. The following effect strength in the sequence was shown by depth. Position had a significant but extremely low effect on all parameters and no effect on n_IT_. The binary interaction effects of the main parameters were also significant but very low.

Depth of cure (DOC, Figure 6) was calculated taking into account the variation in HV with depth and indicating the depth at which HV decreased below 80% of the 0.1 µm sub-surface value. A univariate general linear model identified a significant and moderate effect of the material on the DOC (*p* < 0.001, partial eta-squared η^2^_P_ = 0.363), while position did not significantly influence it (*p* = 0.578). The binary combination of material and position also had no effect (*p* = 0.303). Exp and TPF showed similar (*p* = 0.135) and the significantly highest DOC values (5.3 mm and 5.0 mm, respectively) compared to SDR+ (4.5 mm, *p* < 0.001).

### 3.3. Degree of Cure

The degree of carbon–carbon double bond conversion is shown in Figure 7 as a function of material and depth 300 s post-polymerization. A student *t*-test showed that the differences between the DC at 2 mm and 4 mm depths were statistically similar in all three materials analyzed (*p* = 0.283; *p* = 0.868; *p* = 0.730 for the sequence presented in Figure 7). Even if all three materials are based on methacrylate, a direct comparison of the DC in systems with different chemical compositions of the organic matrix is not relevant to the quality of the polymerization, which is why a direct comparison of the materials with regard to the DC is not provided.

The real-time variation in DC as a function of material and depth is summarized in Figure 8.

The variation in DC over time presented above is best described by an exponential sum function, characterized by a very high correlation factor of R^2^ > 0.96. The first exponential function characterizes the gel phase (parameters “a” and “b”) and the second the glass phase (parameters “c” and “d”; Table 4).

The difference in DC kinetic between the 2 mm and 4 mm depths is very small, with a significantly lower parameter “b” and a significantly higher parameter “c” being observed for Exp at the bottom of the 4 mm samples. All kinetic parameters for the C=C double bond conversion were identical for the 2 and 4 mm depths for SDR+, while for TPF, parameter “a” was significantly lower at the 4 mm depth. In all materials, the DC at the end of the 300 s of real-time observation was statistically the same regardless of depth. The significance of the comparison was assessed taking into account the standard error to calculate the 95% confidence interval, which was the parameter value from which 1.96 × standard error was added and subtracted. Significant differences occur when the calculated confidence intervals do not overlap.

A direct material comparison showed a slightly faster increase in the DC in the gel phase (higher parameter b) for TPF compared to the other two materials, which was also reflected by a slightly higher parameter “a”. Further differences were only observed for the 4 mm depth, with a higher parameter “c” in Exp indicating a slightly more pronounced increase in the glass phase compared to the other materials.

### 3.4. Three-Point Bending Test

One-way ANOVA evidenced a similar and significantly higher flexural strength for Exp and SDR+ (*p* = 0.241) compared to TPF (*p* < 0.001). The mirror constant (A) followed the same trend, but the differences were no longer significant. The flexural modulus (E) behaved in a reverse sequence: TPF > SDR+ > Exp, but this time was clearly differentiated between all materials (*p* < 0.001). The beam deflection followed the above sequence and was clearly distinguished between the materials (*p* < 0.001), while the higher the flexural modulus, the lower the beam deflection (Figure 9a–d, Table 5). In addition, flexural strength data were used to calculate the reliability of the materials using Weibull statistics (Figure 10). High R^2^ values were observed for all groups (R^2^ > 0.90), indicating a very good fit of the Weibull model. A very high and similar reliability (the confidence intervals overlap) was found for Exp and TPF, while the Weibull modulus m was lower for SDR+ (Table 5).

Four fracture patterns were identified, with the absolute highest percentage of failures for all materials being due to volume defects: from 85% for SDR+ and Exp to 90% for TPF. The small remainder of the fractures were identified as surface defects located either at the edge or corner of the fractured sample. One fracture out of sixty was classified as indefinable because the fracture pattern did not allow for identification of the fracture origin (Figure 11).

A significant direct correlation was identified between filler volume and flexural modulus (Pearson coefficient *p* = 0.564) and a corresponding inverse correlation with beam deflection (Pearson coefficient *p* = −0.528). A relatively lower, surprisingly inverse correlation was also found with FS (Pearson coefficient *p* = −0.391).

### 3.5. Light and Scanning Electron Microscopy (SEM) Characterization

Light microscopy: Figure 12a,b shows an example of a fractographic evaluation of the experimental composite Exp, with a volume and a surface defect as the initiator for the fracture.

The scanning electron microscopy (SEM) evaluation is presented in Figure 13. To enable distinction between the structural appearance of the individual components of the filler systems, scanning electron microscopy was performed in electron backscatter diffraction mode. Fillers that contained higher-atomic-order elements in their composition therefore appeared lighter. In addition to compact fillers, both Exp and TPF obviously contained pre-polymerized fillers (PPF). The compact fillers were irregular in shape and larger in TPF and SDR+ than in Exp. Their chemical nature also appeared to differ, with higher-atomic-order elements associated with Exp.

## 4. Discussion

The analyzed experimental bulk-fill flowable composite formulation belongs to the category of novel, universal materials for simplified restoration procedures. The commercial version was launched just a few days ago and is promoted as a material that aesthetically matches a variety of tooth colors, has little shrinkage and can be used in 4 mm increments in the bulk filling technique without capping. The stated depth of cure by curing with an exposure time of 20 s and a modern LCU with high irradiance is clearly confirmed by both the degree of conversion and the depth of cure determined by hardness profiling. In the present study, it was compared with two materials of the same category that have been used successfully in patients for several years [9,25] and were designed to reduce the treatment time for the placement of direct composite restorations. In this context, it should be mentioned that the material SDR was the initiator in the development of flowable bulk-fill materials, which focuses on serving as a flowable lining material on creating enhanced rheological properties that enable material flow into narrow and deep cavities, where condensing of a material is not possible.

A direct comparison between the studied materials is possible by taking into account the kinetic parameters of polymerization, which are best described by an exponential sum function, the first characterizing the gel phase (parameters “a” and “b”) and the second characterizing the glass phase (parameters “c” and “d”). The real-time evaluation of the DC at 2 mm and 4 mm depths shows similar values at the end of the 300 s post-polymerization monitoring period, thus confirming sufficient curing at the bottom of 4 mm increments for all materials. Only in connection with the polymerization kinetic were small differences found. These are hardly noticeable when visualizing the DC curves but are evidenced by the exponential sum function used to adjust the data. These small differences are related in Exp to slightly lower “b” values, the second parameter used to fit the first exponential function that characterizes the gel phase. Since parameter “a” was similar at 2 mm and 4 mm depths, a lower parameter “b” indicates a slightly slower DC increase at the bottom of the 4 mm samples compared to 2 mm during the gel phase, or in other words, the DC at the 4 mm depth reaches the “a” value slightly slower. This behavior is well explained by the fact that the light as it passes through the material is attenuated exponentially with the material thickness, reducing the reaction rate at depth due to a reduction in photons capable of starting the polymerization process. This feature may be reflected by a different molecular architecture at depth compared to the upper regions, involving longer and less cross-linked polymer chains, but without obvious negative effects on mechanical properties. Furthermore, the significantly higher parameter “c” observed for Exp at the bottom of the 4 mm samples, thus the first parameter used to fit the second exponential function to describe the glass phase, compensates for the slightly lower development of the polymerization in the gel phase. These small differences are no longer observed for SDR+, where all kinetic parameters of the C=C double bond conversion were identical for the 2 and 4 mm depths. Characteristic of this material is the composition of the organic matrix, which contains high-molecular-weight urethane dimethacrylate monomers to reduce shrinkage. The course of the DC curves over the exposure time shows that Exp and SDR+ are identical in the first ca. 7 s. After the initial steep gradient, the curve flattens out, starting significantly faster in SDR than in Exp. This could well be related to the faster restriction of monomer mobility, which occurs earlier with the significantly larger monomers in SDR+. The same comparison clearly shows that TPF has a slightly steeper curve with an initially higher conversion rate of the C=C double bonds, which is reflected by the highest “a” and “b” parameters in the material comparison. This can be associated with the additional use of a tailored germanium-based photo-initiator, Ivocerin [14]. Two germanium initiators [bis-(4-methoxybenzoyl)diethylgermane] (BTMGe and DBDEGe) are described in the literature, which have absorption maxima at wavelengths (λ_max_ = 411 nm and λ_max_ = 418 nm) smaller than CQ (λ_max_ = 468 nm) [14]. These are Norrish type I photo-initiators that can generate two radicals through alpha cleavage when exposed to light and do not require additional co-initiators. The reactivity of these initiators is considered to be higher compared to CQ, which is due to the higher molar extinction coefficient (ε_λ_) (ε_λ_ = measure of the electromagnetic radiation that an initiator absorbs at a certain wavelength) [14]. A high ε_λ_ indicates a high probability of light absorption at a particular wavelength. Another advantage that should be mentioned is that the color stability of composites with germanium initiators is rated as very good [26]. The description of the initiator shows that in the present study design, the blue LED LCU used is not ideally adapted to Ivocerin. A small adjustment to the above statement needs to be done, as the absorption spectrum of Ivocerin extends slightly into the blue wavelength range [14] and can therefore be assumed to contribute to improving polymerization. It is also worth mentioning that the exposure time for all materials was 20 s, longer than the manufacturer stated for SDR+ (10 s), so the disadvantage should be compensated for.

Similar to the DC, the DOC also confirms that the curing of the materials in at least 4 mm-thick increments is sufficient. The evaluation according to the current standards even yields slightly higher values for the thickness of the increments that are cured adequately, which are between 4.5 and 5.3 mm. However, it should be noted that ideal polymerization conditions were created in the study, so the clinical indication should not exceed the 4 mm mark.

Apart from the quality of the polymerization in depth, the examined materials show some differences in the measured elastoplastic behavior, which are well explained by the filler content (Table 1). In this context, the experimental material show a slightly lower hardness and elastic modulus due to its lower filler content. The corresponding higher polymer matrix content in the experimental material manifests in a higher elastic and total indentation work. Compared with the literature data measured under similar conditions, the experimental material can be classified in the range of materials of the same category, but, similar to the materials used for comparison in the present study, does not yet approach values measured in the human dentin, especially for the indentation modulus [3]. These observations, also confirmed by the low elastic modulus measured in the three-point bending test on a macroscopic scale, support the statement that flowable bulk materials in larger cavities should only be used as a lining and supported by a covering material with a higher elastic modulus. Interestingly, the creep is not affected by the lower filler content in the experimental material, which can be related to the presence of relatively higher nano-filler amount, as observed in the SEM analysis. TPF and Exp both contained pre-polymer fillers in addition to the compact fillers. Since the SEM measurements were performed in electron backscatter diffraction mode, differences in the composition of the compact fillers became evident, as they appear much lighter in Exp, thus indicating a composition of higher-atomic-order elements. Such undertakings are mostly performed to enhance the radiopacity of the materials, but come at the expense of light transmittance, since the match between the refractive index of the fillers and organic matrix is important [27]. The greater this difference, the more light is scattered at the interface between filler and matrix and therefore lost in the deeper layers [28,29]. While silicon dioxide fillers and alkaline glasses have a comparable refractive index (1.46–1.5) to traditional monomer blends (≈1.5), metal oxides such as aluminum oxide (Al_2_O_3_), titanium oxides (TiO_2_), zinc oxide (ZnO), or zirconium oxide (ZrO_2_) are characterized by higher refractive indices (2.00–2.16). Although the type of filler is not clearly stated, the increased opacity was obviously taken into account as the manufacturer’s indicated exposure time for Exp is 20 s, as opposed to the 10 s indicated for the other two materials. On a different note, the depth profiling of the mechanical properties for all analyzed materials shows an increase in HV, HM, and E_IT_ up to a depth of 1–1.5 mm, followed by a constant or slightly decreasing trend. This phenomenon is not due to fitting the data to create the mean curves, as it is observed on each individual sample and does not depend on the LCU used [17]. In fact, this can be explained as an insulating effect of the upper layer that helps to improve curing beneath, because polymerization occurs exothermally and the surface cools faster than the core. This behavior can be observed above all in flowable composites, i.e., materials with a higher polymer content.

Finally, the materials were analyzed in regard to their macroscopic mechanical behavior. The flexural strength data were well in excess of the ISO standard requirements but must be taken into account with caution for flowable materials as they are related to the higher flexibility of the beams during testing and therefore the longer time to failure. The modulus of elasticity is therefore an indispensable companion in such assessments, as it is directly related to the amount of inorganic filler. In fact, it turned out to be low for all three materials, but is one of the least measured parameters that allow for clear distinction between all three materials. Since two of the tested materials contain a pre-polymerized filler, the total amount of inorganic filler cannot be clearly distinguished since this type of filler also contains a polymer. However, the elastic modulus can be considered as a predictable parameter for the amount of inorganic filler in dental composites and decreases in the sequence TPF > SDR+ > Exp. In addition, the reliability of the materials expressed by the Weibull analysis is impressively high, which is certainly due to the high adaptability of the material to the mold during specimen preparation and therefore to the creation of samples with fewer critical defects able to initiate fracture. This was also reflected by the impressively high number of fractures initiated from volume defects. The lower reliability in SDR+ may be related to the larger size of the inorganic filler, which creates large defects after being pooled out, apparently large enough to create critical defects that can initiate fracture. At n = 20, the number of test specimens for the three-point bending test was already four-fold higher than specified in ISO 4049, but this number is still at the lower limit for the Weibull analysis. An even higher number of test specimens would further refine the results.

## 5. Conclusions

The experimental bulk flow composite meets the requirements related to its intended use. It can be cured in 4 mm increments, which is confirmed by both the depth of cure and the constancy of the degree of cure at 2 mm and 4 mm depths. It also exceeds ISO standard requirements for flexural strength. The comparison with clinically established materials shows similar or better flexural strength, reliability, and depth of cure. In contrast, the hardness (Vickers and Martens) and indentation modulus are slightly lower, which is well related to the lower amount of filler and the corresponding higher beam deflection in the three-point bending test. The creep is comparable to the materials examined, while the elastic deformation during indentation and the total indentation work are higher. Since the modulus of elasticity measured on macroscopic and microscopic scales is low for all flowable bulk materials, they should only be used as a lining in larger cavities and supported by a cover material with a sufficient modulus of elasticity.

## Figures and Tables

**Figure 1 bioengineering-11-00159-f001:**
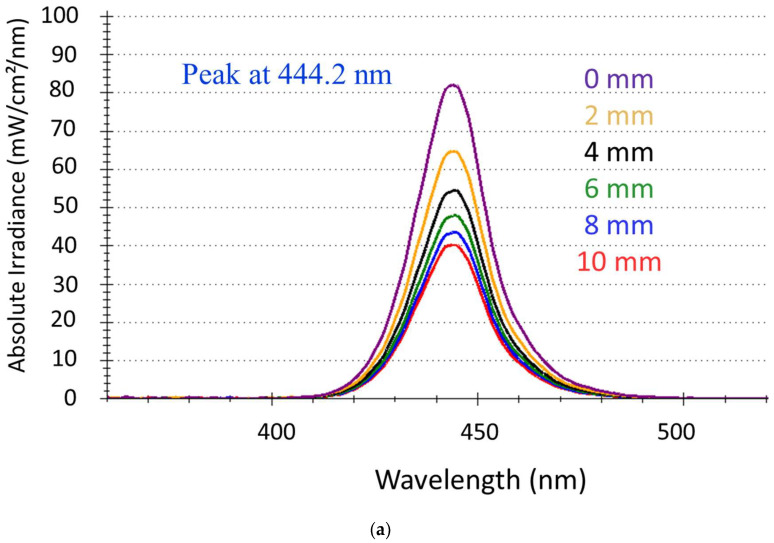
LCU characteristics as a function of exposure distance: (**a**) spectral distribution; (**b**) variation in the irradiance within the 20 s exposure; (**c**) exponential decay of irradiance with exposure distance.

**Figure 2 bioengineering-11-00159-f002:**
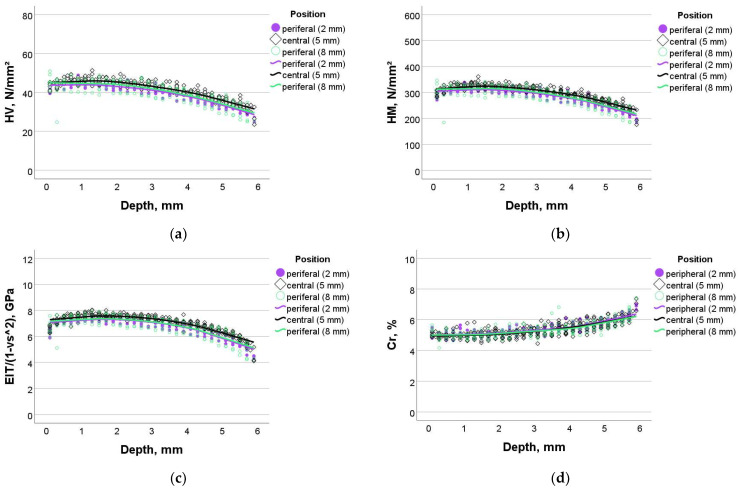
Dependence of the measured parameters on sample depth and sample width in the experimental composite (Exp): (**a**) Vickers hardness, (**b**) Martens hardness, (**c**) indentation modulus, and (**d**) creep.

**Figure 3 bioengineering-11-00159-f003:**
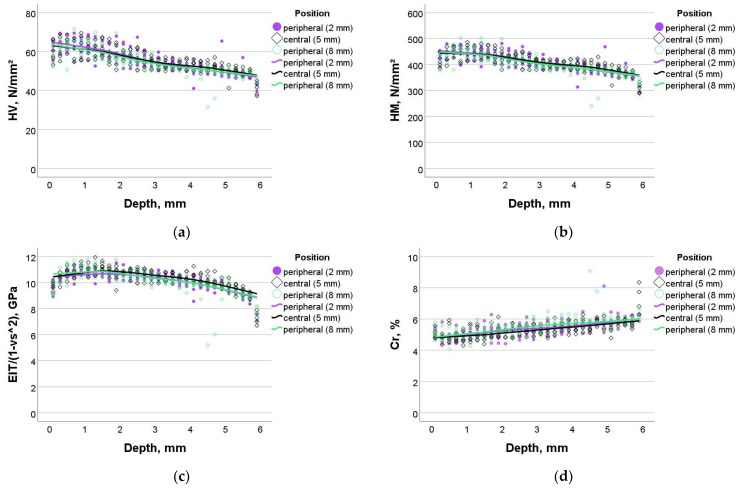
Dependence of the measured parameters on sample depth and sample width in SDR+: (**a**) Vickers hardness, (**b**) Martens hardness, (**c**) indentation modulus, and (**d**) creep.

**Figure 4 bioengineering-11-00159-f004:**
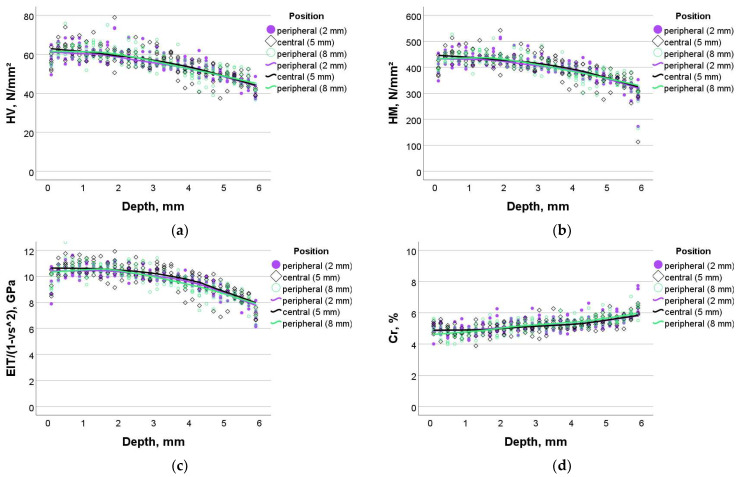
Dependence of the measured parameters on sample depth and sample width in TPF: (**a**) Vickers hardness, (**b**) Martens hardness, (**c**) indentation modulus, and (**d**) creep.

**Figure 5 bioengineering-11-00159-f005:**
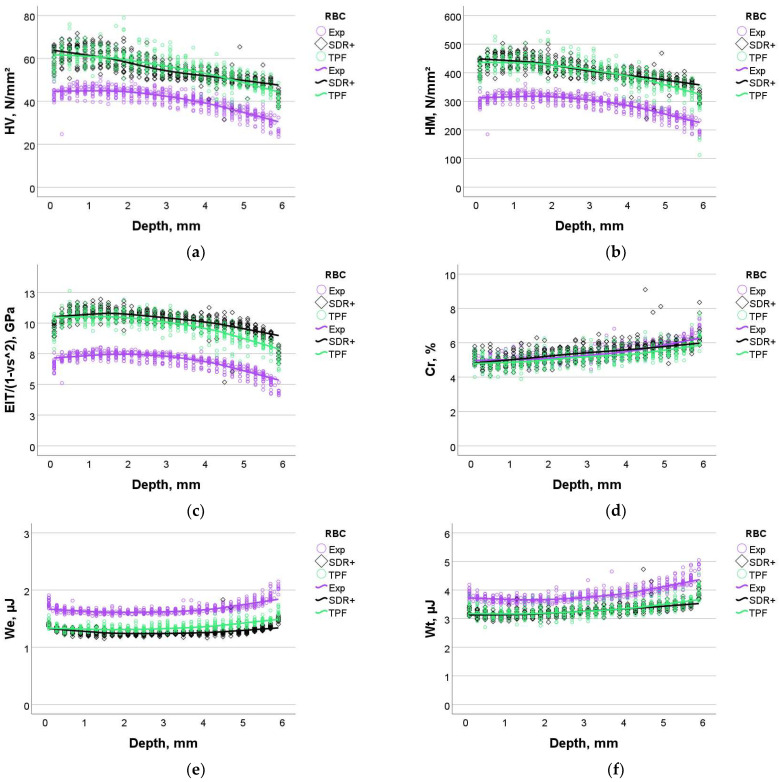
Material comparison: dependence of the measured parameters with depth: (**a**) Vickers hardness, (**b**) Martens hardness, (**c**) indentation modulus, (**d**) creep, (**e**) elastic reverse deformation work of indentation (W_e_), and (**f**) total mechanical work of indentation (W_t_).

**Figure 6 bioengineering-11-00159-f006:**
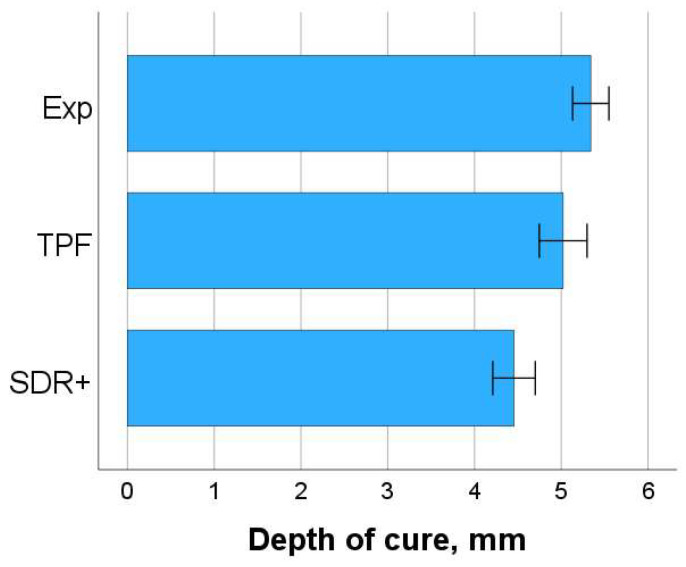
Depth of cure (DOC, mm) of the analyzed composites. Since measurement position (central or peripheral) had no influence on DOC, data were pooled for each material. Mean values with 95% confidence interval are given.

**Figure 7 bioengineering-11-00159-f007:**
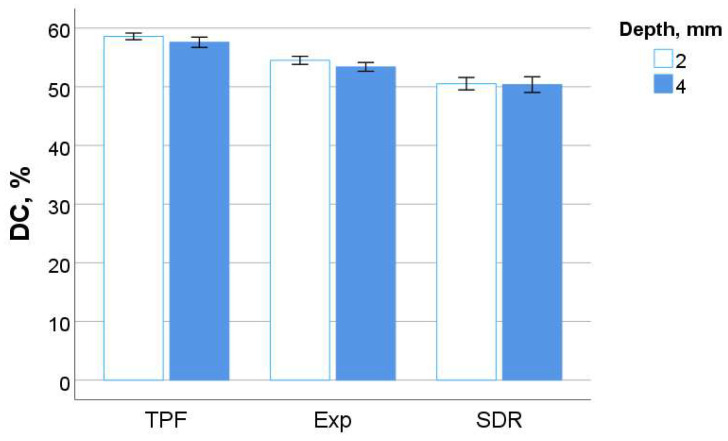
Degree of cure as a function of material and sample depth: mean values with 95% confidence interval.

**Figure 8 bioengineering-11-00159-f008:**
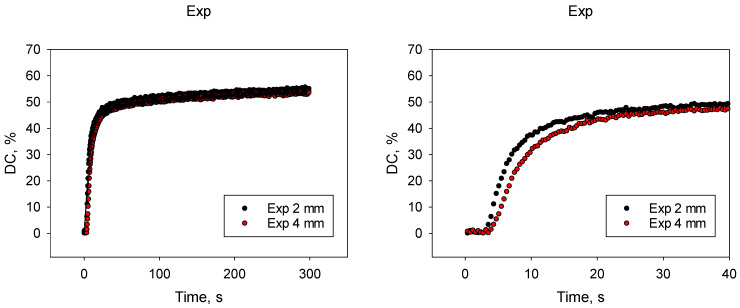
Real-time variation in the degree of conversion (DC) depending on time (up to 300 s), depth (2 mm and 4 mm) and material (mean values, n = 6). A close-up is shown on the right side for each material and includes the exposure time (20 s) and an additional 20 s to illustrate differences within each material between curing at 2 mm and 4 mm depths. The direct material comparison is provided for the 2 mm depth.

**Figure 9 bioengineering-11-00159-f009:**
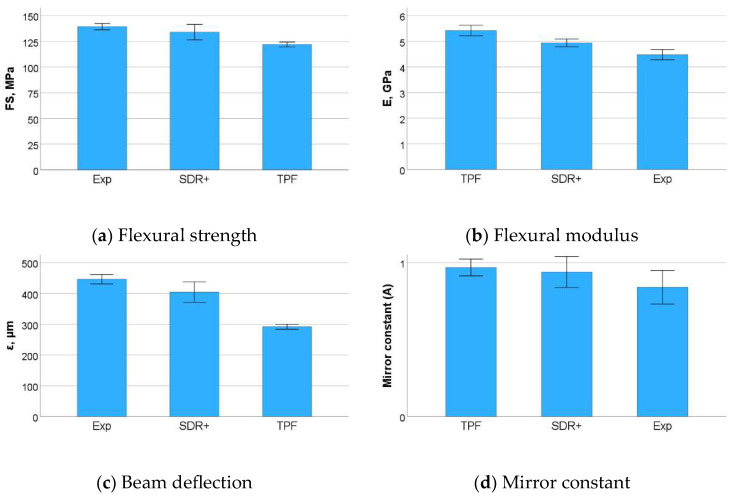
Outcome of the three-point bending test: (**a**) flexural strength (FS); (**b**) flexural modulus; (**c**) beam deflection, and (**d**) mirror constant. Presented data are mean values with 95% confidence interval.

**Figure 10 bioengineering-11-00159-f010:**
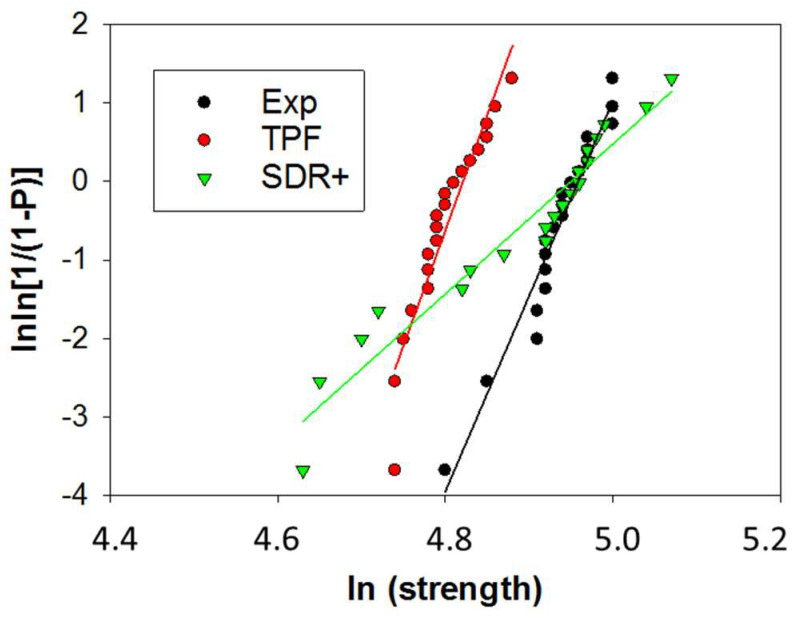
Weibull diagram representing the empirical cumulative distribution function of the strength data in the analyzed composites. Linear regression was used to numerically evaluate the goodness of fit and estimate the parameters of the Weibull distribution. Weibull parameters (m) and coefficient of determination of the regression model (R^2^) are summarized in Table 5.

**Figure 11 bioengineering-11-00159-f011:**
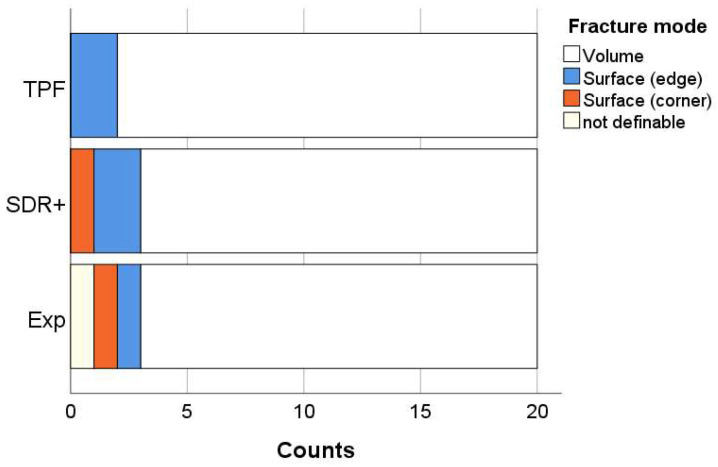
Classification of fracture patterns for each of the investigated composites.

**Figure 12 bioengineering-11-00159-f012:**
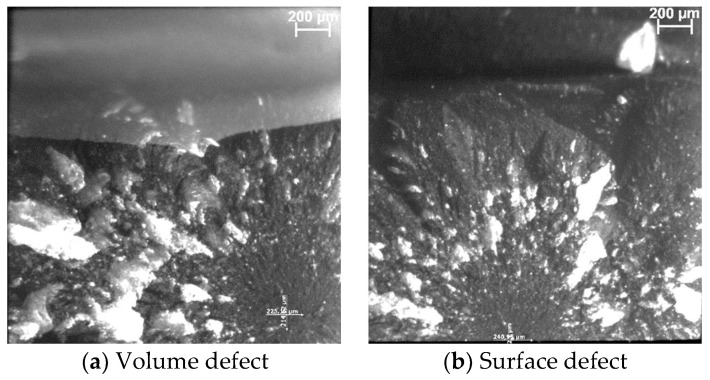
Fractographic evaluation of the three-point bending fractured samples exemplified for Exp. (**a**) The fracture was initiated from a sub-surface (volume) defect and (**b**) from a surface defect.

**Figure 13 bioengineering-11-00159-f013:**
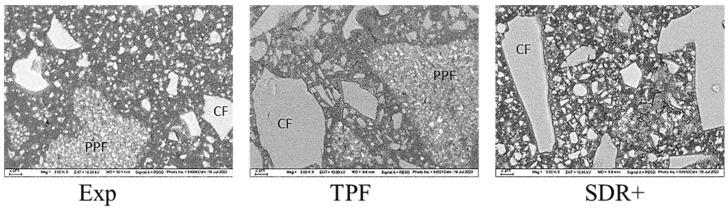
SEM images: structural appearance of the filler systems, measured in the electron backscatter diffraction mode (abbreviation: PPF = pre-polymerized fillers; CF = compact fillers).

**Table 1 bioengineering-11-00159-t001:** Analyzed RBCs: Abbreviation (code), brand, manufacturer, LOT, and composition, as indicated by the manufacturer. Exposure time was 20 s in all materials (n.s. = not specified).

Code	Brand	Manufacturer	LOT	Filler wt./vol.%
Exp	Brilliant BulkFill Flow	Coltene	M38595	56.0/38.5
SDR+	SDR flow+ universal	Dentsply Sirona	2206000502	n.s./47.3
TPF	Tetric PowerFlow, ^IV^ A	Ivoclar Vivadent	Z03W68	68.2/46.4

**Table 2 bioengineering-11-00159-t002:** Descriptive statistic for IIT parameters: Martens and Vickers hardness (HM, HV, N/mm^2^), indentation modulus (E_IT_, GPa), creep (Cr, %), elastic reverse deformation work of indentation (W_e_), total mechanical work of indentation (W_t_) (µJ), and their relationship (n_IT_). Mean and standard deviation (SD) are given. Similar letters indicate homogeneous groups; Tukey’s post-hoc test (α = 0.05).

Parameter	RBC	0.1 mm		2 mm		4 mm	
		Mean	SD	Mean	SD	Mean	SD
HM	Exp	291.5 a	26.9	294.1 a	21.7	314.5 a	12.9
	SDR+	439.5 b	25.1	396.3 a	12.7	427.7 b	21.3
	TPF	422.0 b	35.6	401.8 b	23.3	434.3 b	35.6
HV	Exp	41.9 a	4.3	40.8 a	3.5	44.1 a	2.1
	SDR+	63.1 b	4.2	52.6 b	2.0	58.2 b	4.2
	TPF	60.0 b	5.0	55.2 c	3.7	60.6 b	6.1
E_IT_	Exp	6.6 a	0.5	7.0 a	0.4	7.4 a	0.4
	SDR+	10.1 b	0.5	10.2 c	0.3	10.8 b	0.4
	TPF	10.1 b	0.9	9.7 b	0.4	10.5 b	0.6
Cr	Exp	5.0 a	0.3	5.4 a	0.4	5.1 a	0.4
	SDR+	4.9 a	0.4	5.6 a	0.3	5.2 a	0.5
	TPF	4.9 a	0.4	5.5 a	0.4	5.0 a	0.4
We	Exp	1.7 a	0.1	1.7 c	0.1	1.6 c	0.0
	SDR+	1.4 b	0.0	1.2 a	0.0	1.2 a	0.0
	TPF	1.4 b	0.1	1.4 b	0.0	1.3 b	0.0
Wt	Exp	3.8 a	0.1	3.9 a	0.2	3.7 b	0.1
	SDR+	3.2 b	0.1	3.3 b	0.1	3.1 a	0.1
	TPF	3.2 b	0.2	3.4 b	0.2	3.2 a	0.2
n_IT_	Exp	45.5 a	1.1	42.8 c	1.1	43.9 c	0.8
	SDR+	43.1 b	1.4	37.6 a	1.3	39.5 a	1.8
	TPF	42.4 b	1.8	40.4 b	2.0	41.1 b	1.7

**Table 3 bioengineering-11-00159-t003:** Multivariate analysis (general linear model) to evaluate the effect strength of the main parameters—RBC, filler volume %, position, and depth—as well as their binary interaction on the measured parameters. Partial eta-squared values are given; n.s. = not significant.

Parameter	HM	HV	E_IT_	n_IT_	W_e_	W_t_	Cr
RBC	0.876	0.827	0.921	0.649	0.928	0.777	0.065
Filler vol%	0.670	0.575	0.750	0.532	0.814	0.544	0.026
Position	0.025	0.019	0.052	n.s.	0.042	0.029	0.014
Depth	0.690	0.691	0.721	0.334	0.604	0.599	0.509
RBC × Position	0.023	0.026	0.010	0.015	0.012	0.016	0.007
RBC × Depth	0.117	0.131	0.113	0.113	0.167	0.096	0.062
Depth × Position	0.075	0.073	0.101	0.057	0.087	0.085	0.074

**Table 4 bioengineering-11-00159-t004:** Parameters of the exponential sum function used to fit the real-time variation in the degree of conversion (DC) as a function of material and depth. Coefficient of determination (R^2^) and the mean and standard error (Std. Error) for the calculated parameters are given.

RBC	Depth	R^2^		a	b	c	d
Exp	2 mm	0.96	mean	48.69	0.12	7.97	0.00
			Std. Error	0.35	0.00	0.69	0.00
	4 mm	0.96	mean	48.23	0.09	12.62	0.00
			Std. Error	0.37	0.00	0.61	0.00
SDR+	2 mm	0.97	mean	43.01	0.12	9.25	0.01
			Std. Error	0.33	0.00	0.34	0.00
	4 mm	0.97	mean	44.19	0.11	9.17	0.00
			Std. Error	0.29	0.00	0.77	0.00
TPF	2 mm	0.96	mean	53.11	0.15	7.37	0.00
			Std. Error	0.30	0.00	0.83	0.00
	4 mm	0.96	mean	51.20	0.15	8.22	0.01
			Std. Error	0.34	0.00	0.65	0.00

**Table 5 bioengineering-11-00159-t005:** Variation in measured parameters in the three-point bending test as a function of material. Within one parameter, values denoted by the same superscript are statistically similar (SE = standard error; SD = standard deviation).

COD	FS [MPa] Mean + SD		Weibull Parameters m, SE, R^2^			A [MPa√m] Mean + SD		E [GPa] Mean + SD		ε [µm] Mean + SD	
Exp	139.3 ^a^	6.6	24.9 ^a^	1.4	0.95	0.8 ^a^	0.2	4.5 ^a^	0.4	446.0 ^a^	32.7
SDR+	134.0 ^a^	16.0	9.5 ^b^	0.6	0.94	0.9 ^a^	0.2	4.9 ^b^	0.3	404.1 ^b^	70.6
TPF	122.0 ^b^	5.0	29.3 ^a^	2.3	0.90	1.0 ^a^	0.1	5.4 ^c^	0.4	291.6 ^c^	17.6

## Data Availability

Data are contained within the article.

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
