# Peer review of "Profiling Elastoplastic and Chemical Parameters to Assess Polymerization Quality in Flowable Bulk-Fill Composites"

_bioengineering, 2024, doi:10.3390/bioengineering11020159_

Round 1

Reviewer 1 Report

Comments and Suggestions for Authors

This manuscript is written clearly and intelligibly.The content of this manuscript will be of some interest to readers.

Author Response

All comments to the corresponding author have been addressed independently below. The authors’ rebuttal is always in BLUE and where changes have been added to the revised manuscript in light of the reviewer's comments these are presented in RED.

The author would firstly like to thank the reviewers for taking the time to read and critically appraise the manuscript and secondly to thank the reviewers for their positive constructive comments in improving the work.

Comments and Suggestions for Authors

Reviewer 1

This manuscript is written clearly and intelligibly.The content of this manuscript will be of some interest to readers.

Author’s response:  I am grateful for this appreciation. Thank you.

Reviewer 2 Report

Comments and Suggestions for Authors

The paper shows interesting research related to polymerization quality analysis. The manuscript is well-structured and well-written. However, there are some minor issues that need some improvements:

1. The abstract part needs to be rewritten. Now it provides some general information about the paper, when it should describe the aim of this work and most important outcomes supported by quantified values.

2. 17 citations is not a big amount of literature, especially, since this topic is not so novel. What is more, there is a lack of a proper literature review that is directly related to the main topic of this work. 

3. The aim of this work is not clear, please make it more understandable, and put its proper background. 

4. Figures 2-5 are invisible (especially curves, and points).

Author Response

All comments to the corresponding author have been addressed independently below. The authors’ rebuttal is always in BLUE and where changes have been added to the revised manuscript in light of the reviewer's comments these are presented in RED.

The author would firstly like to thank the reviewers for taking the time to read and critically appraise the manuscript and secondly to thank the reviewers for their positive constructive comments in improving the work.

Reviewer 2

The paper shows interesting research related to polymerization quality analysis. The manuscript is well-structured and well-written. However, there are some minor issues that need some improvements:

Author’s response: Thank you for the appreciation.

  1. The abstract part needs to be rewritten. Now it provides some general information about the paper, when it should describe the aim of this work and most important outcomes supported by quantified values.

Author’s response: I agree with your comment – ​​the abstract has been rewritten, please see the revised manuscript.

  1. 17 citations is not a big amount of literature, especially, since this topic is not so novel. What is more, there is a lack of a proper literature review that is directly related to the main topic of this work. 

Author’s response:  The introduction chapter briefly presents the material category, the most important advantages and disadvantages of bulk filling, and the difference between low and high-viscosity materials. In addition, the chemical composition and chronological development are discussed, with the problem of translucency and how it was resolved being highlighted. I then discussed new developments, emphasizing that not only classic methacrylate, but also Ormocere and new polymerization techniques (RAFT) etc. have been implemented, to then focus the presentation on the progressive new development and rationality of developing new materials. In my opinion, I have touched on the most important aspects related to the material category. Please note that an introduction is not a review and it aims to briefly present the problems and rationality of conducting the study that follows in the narrative. Based on the addressed criticism, I slightly extended the introduction by mentioning the deficits of inadequate polymerization and rewrote the aim of the study in this context.

  1. The aim of this work is not clear, please make it more understandable, and put its proper background. 

Author’s response:  The end of the introduction specifies that applying a composite in layers of up to 4-5 mm entails the increasing risk that not enough light will reach the bottom of the restoration and therefore insufficient polymerization may occur in deeper layers, and then new developments need for careful and thorough material characterization from physicochemical and mechanical perspectives.

For better clarity, I have deepened the sentence regarding the aim of the study. Thank you for pointing out this weakness.

  1. Figures 2-5 are invisible (especially curves, and points).

Author’s response:  I would like to thank you for this observation. The graphics 2-5 have been changed accordingly and more contrast has now been created between the displayed parameters and materials.

Reviewer 3 Report

Comments and Suggestions for Authors

The article "Profiling elastoplastic and chemical parameters to assess polymerization quality in flowable bulk-fill composites" by Nicoleta Ilie et al. focuses on the development of polymer-based composite materials, specifically bulk-fill composites. The study aims to assess the quality of curing in these materials by analyzing their elastoplastic behavior, degree of curing, polymerization kinetics, and mechanical properties. The methods include instrumented indentation tests, Fourier Transform Infrared (FTIR) spectroscopy, three-point bending tests, fractography, and reliability analysis. The study also examines the use of light-curing units and their impact on the polymerization process.

Clarity and Structure: The article is well-structured, with clear sections. However, it could benefit from a more concise introduction, focusing on the key research question and its significance in the field of bioengineering.

Methodology: The methods used are appropriate for the study's goals. However, the article could provide more details on the selection criteria for the materials tested. The statistical analysis methods are adequately described, but It would be beneficial if the study could include a discussion on the limitations of these methods and how they might affect the results.

Results and Discussion: The results are comprehensive and well-presented. However, the discussion could be enhanced by comparing findings with previous studies more extensively to contextualize the results within the existing body of research. The implications of the findings for practical applications in dentistry should be elaborated upon.

Figures and Tables:  Ensure that all figures and tables are clearly labeled and referenced within the text. Consider adding more graphical representations of data to enhance readability and comprehension.

 Conclusion: The conclusion is coherent with the study's findings but could be strengthened by highlighting future research directions and potential applications of the research.

References: The reference list appears comprehensive.

Language and Grammar: The paper would benefit from proofreading to correct minor language and grammatical errors, enhancing its overall readability.

Comments on the Quality of English Language

Minor editing of English language required

Author Response

All comments to the corresponding author have been addressed independently below. The authors’ rebuttal is always in BLUE and where changes have been added to the revised manuscript in light of the reviewer's comments these are presented in RED.

The author would firstly like to thank the reviewers for taking the time to read and critically appraise the manuscript and secondly to thank the reviewers for their positive constructive comments in improving the work.

Reviewer 3

The article "Profiling elastoplastic and chemical parameters to assess polymerization quality in flowable bulk-fill composites" by Nicoleta Ilie et al. focuses on the development of polymer-based composite materials, specifically bulk-fill composites. The study aims to assess the quality of curing in these materials by analyzing their elastoplastic behavior, degree of curing, polymerization kinetics, and mechanical properties. The methods include instrumented indentation tests, Fourier Transform Infrared (FTIR) spectroscopy, three-point bending tests, fractography, and reliability analysis. The study also examines the use of light-curing units and their impact on the polymerization process.

Clarity and Structure: The article is well-structured, with clear sections. However, it could benefit from a more concise introduction, focusing on the key research question and its significance in the field of bioengineering.

Author’s response:  Thank you for your comment. I tried to keep the introduction as concise as possible, but still addressing the most important aspects of the material category. It's always a personal perspective, which may seem too brief to one person (reviewer 2) and too detailed to another (reviewer 3). I also added a phrase to the significance of adequate curing. In addition, I rewrote the aim of the study.

Methodology: The methods used are appropriate for the study's goals. However, the article could provide more details on the selection criteria for the materials tested. The statistical analysis methods are adequately described, but It would be beneficial if the study could include a discussion on the limitations of these methods and how they might affect the results.

Author’s response:  The selection criteria for the materials used to compare the experimental materials were based on their clinical effectiveness, which has been studied over the years. I added a sentence about it in the revision. Also, the limitation of the statistics is expressed at the end of the discussion.

Results and Discussion: The results are comprehensive and well-presented. However, the discussion could be enhanced by comparing findings with previous studies more extensively to contextualize the results within the existing body of research. The implications of the findings for practical applications in dentistry should be elaborated upon.

Author’s response:  Thank you for the pertinent remarks, which are now incorporated in the revised discussion.

Figures and Tables:  Ensure that all figures and tables are clearly labeled and referenced within the text. Consider adding more graphical representations of data to enhance readability and comprehension.

Author’s response:  The figures have been modified accordingly and more contrast has now been created between the displayed parameters and materials.

 Conclusion: The conclusion is coherent with the study's findings but could be strengthened by highlighting future research directions and potential applications of the research.

Author’s response:  Since the study aimed to place an experimental material in the ranks of clinically successful materials, I fear that any future research directions are too speculative. The limitations of the material in its clinical application have been addressed in the revised manuscript.

References: The reference list appears comprehensive.

Author’s response:  I am grateful for this appreciation. Thank you.

Language and Grammar: The paper would benefit from proofreading to correct minor language and grammatical errors, enhancing its overall readability.

Author’s response:  Language and Grammar were checked again and some errors were corrected.

Reviewer 4 Report

Comments and Suggestions for Authors

In this contribution, the author assessed the polymerization quality of bulk-fill composites by profiling the elastoplastic performance, degree of cure, and polymerization kinetics. Those systematic characterizations are inspiring to the evaluation of cure quality in millimeter increments, and this work is suitable for the readership of Bioengineering. Therefore, I would recommend its publication after addressing the following comments.

1. What do the gel and glass phases correlate to (Line 343), i.e., is the glass phase cured or crystalline polymer?

2. Why do HW and HV increase within 1-2 mm depth in Figure 2, though the irradiance decreases monotonically in Figure 1?

3. How are the pre-polymerized fillers distinguished in the SEM images (Line 411 and Figure 13)?

4. Although the title highlights chemical parameters, this research does not provide much insight into chemistry. Besides, how is the polymerization quality of commercial bulk-fill composites previously or routinely assessed? Adding this content to the Introduction enriches the background and highlights the novelty of this manuscript.

5. Please define RBCs (Line 68), HV, HW, Cr (Line 252), and COD (Table 1). Besides, Table 2 needs corrections of subscripts.

Author Response

All comments to the corresponding author have been addressed independently below. The authors’ rebuttal is always in BLUE and where changes have been added to the revised manuscript in light of the reviewer's comments these are presented in RED.

The author would firstly like to thank the reviewers for taking the time to read and critically appraise the manuscript and secondly to thank the reviewers for their positive constructive comments in improving the work.

Reviewer 4

In this contribution, the author assessed the polymerization quality of bulk-fill composites by profiling the elastoplastic performance, degree of cure, and polymerization kinetics. Those systematic characterizations are inspiring to the evaluation of cure quality in millimeter increments, and this work is suitable for the readership of Bioengineering.

Therefore, I would recommend its publication after addressing the following comments.

  1. What do the gel and glass phases correlate to (Line 343), i.e., is the glass phase cured or crystalline polymer?

Author’s response:  Methacrylate and PMMA are thermoplastic, amorphous polymers. During the polymerization process, the propagation stage is characterized by the growth of the polymer chain and is divided into three phases: the quasistatic process, the gel phase, and the glass phase. After normal chain growth (quasistatic process), the number of high-molecular-weight chains increases, and thus also the viscosity of the reaction solution. The reaction rate increases compared with the quasistatic process in the gel phase (= the Trommsdorff–Norrish effect) as the number of terminations decreases as a result of a viscosity increase in the reaction solution. At further progress of the polymerization process, the reaction solution turns into a gel-like state and the reaction rate decreases (glass effect). Since the propagation phases of the polymerization process are described by different subphases, it was previously shown that the decrease of C–C double bonds during polymerization, measured by Fourier transform infrared (FTIR) spectroscopy, can be described by a sum of two exponential functions, each of them characterizing one subphase.

I have therefore cited the reference to materials and methods in the revision.

  1. Why do HW and HV increase within 1-2 mm depth in Figure 2, though the irradiance decreases monotonically in Figure 1?

Author’s response:  I really appreciate the reviewer's attentive analysis. This phenomenon is less described in the literature. When the mechanical properties are precisely profiled, the increase in HV, HV, or E can be observed primarily in flowable composites, i.e. materials with a higher polymer content. This is most probably an insulating effect and not an effect of the LCU, as the polymerization occurs exothermally and the surface cools faster than the core. I added a note in the revised discussion on the topic.

  1. How are the pre-polymerized fillers distinguished in the SEM images (Line 411 and Figure 13)?

Author’s response:  The images were labeled accordingly in the revision to recognize the difference between PPF and compact fillers. Thanks for the tip.

  1. Although the title highlights chemical parameters, this research does not provide much insight into chemistry. Besides, how is the polymerization quality of commercial bulk-fill composites previously or routinely assessed? Adding this content to the Introduction enriches the background and highlights the novelty of this manuscript.

Author’s response:  Thank you for the observation. FTIR and mechanical properties are in fact the most accepted methods for characterizing the curing quality of dental materials.

  1. Please define RBCs (Line 68), HV, HW, Cr (Line 252), and COD (Table 1). Besides, Table 2 needs corrections of subscripts.

Author’s response:  Thank you for this pertinent observation. Abbreviations have been explained. The superscripts have obviously been modified by the change to the journal format. For this reason, I removed “superscripts” from the table heading.